# Patterns of Wolf Dispersal Respond to Harvest Density across an Island Complex

**DOI:** 10.3390/ani14040622

**Published:** 2024-02-15

**Authors:** Gretchen H. Roffler, Kristine L. Pilgrim, Benjamin C. Williams

**Affiliations:** 1Alaska Department of Fish and Game, Division of Wildlife Conservation, Douglas, AK 99824, USA; 2National Genomics Center for Wildlife and Fish Conservation, Rocky Mountain Research Station, USDA Forest Service, Missoula, MT 59802, USA; kristine.pilgrim@usda.gov; 3Auke Bay Laboratories, Alaska Fisheries Science Center, National Marine Fisheries Service, National Oceanic and Atmospheric Administration, Juneau, AK 99801, USA; ben.williams@noaa.gov

**Keywords:** *Canis lupus*, dispersal, islands, noninvasive genetic monitoring

## Abstract

**Simple Summary:**

Dispersal is a regular part of the life cycle of most wolves, although patterns of dispersal are variable and may be affected by factors such as wolf densities, prey availability, and human-caused mortality. Because dispersal links the network of wolf packs and is the mechanism for recolonizing vacant territories, a better understanding of wolf dispersal can help predict population viability and inform management. We used genetic data collected over a decade to better understand wolf dispersal patterns in the Prince of Wales Island complex in southeast Alaska and revealed a higher degree of population connectivity across the islands than previously known. Annual dispersal rates ranged from 9 to 23% and had a weakly positive relationship with wolf density. Wolves dispersed 41.9 km on average (SD = 23.7 km), and males and females did not disperse at different rates. Wolves dispersed both to and from the small islands in the complex and the larger Prince of Wales Island, indicating bidirectionality of movement. Dispersal patterns revealed the need to account for regional population structure in wolf management strategies in this island archipelago.

**Abstract:**

Wolves are highly mobile predators and can disperse across a variety of habitats and over long distances. However, less is known about dispersal capabilities across water and among islands. The biogeography of island systems fosters spatially structured local populations, and their degree of connectivity may influence the dynamics and long-term viability of the regional population. We sought to quantify wolf dispersal rate, distance, and dispersal sex bias throughout Prince of Wales Island, a 6670 km^2^ island in southeast Alaska, and the surrounding islands that constitute the wildlife management unit (9025 km^2^). We also investigated patterns of dispersal in relation to hunting and trapping intensity and wolf population density. We used DNA data collected during 2012–2021 long-term monitoring efforts and genotyped 811 wolves, 144 of which (18%) were dispersers. Annual dispersal rates were 9–23% and had a weakly positive relationship with wolf density. Wolves dispersed 41.9 km on average (SD = 23.7 km), and males and females did not disperse at different rates. Of the dispersing wolves, 107 died, and the majority (n = 81) died before they were able to settle. The leading manner of death was trapping (97% of mortalities), and wolves tended to disperse from areas with low harvest density to areas where harvest density was relatively higher. Dispersal occurred both to and from small islands and the larger Prince of Wales Island, indicating bidirectional as opposed to asymmetrical movement, and the genetic overlap of wolf groups demonstrates connectivity throughout this naturally patchy system. Island ecosystems have different predator–prey dynamics and recolonization processes than large, intact systems due to their isolation and restricted sizes; thus, a better understanding of the degree of population connectivity including dispersal patterns among islands in the Prince of Wales archipelago could help inform the management and research strategies of these wolves.

## 1. Introduction

Dispersal is a key mechanism for population persistence in habitats that are subdivided due to natural or anthropogenic fragmentation [1]. The success of colonization from occupied patches within the metapopulation network will depend on distance from the colonizing source, and the ability of the organism to disperse among habitat patches, including across potential barriers [2,3]. A further impediment to dispersal is the increased risk incurred by the individual moving through unfamiliar territory and exposed to hazards such as predators, territorial defense from conspecifics, and anthropogenic threats such as hunting and trapping [4,5]. In addition, dispersal behavior is energetically expensive, which is exacerbated by unknown access to resources in route or at the destination the individual ultimately occupies if successful [6].

The mainland and island archipelagos of coastal southeast Alaska (USA) and British Columbia (Canada) are a system naturally fragmented by islands, steep topography, and glaciers [7]. Differential colonization has shaped the distribution of terrestrial mammals throughout this region due to isolation [8] and access following glacial retreat [9], resulting in a high concentration of endemic species and variable species assemblages throughout the islands and mainland [10]. Gray wolves (*Canis lupus*), herein referred to as wolves, are a prominent predator in this region and occur throughout the mainland, many of the larger islands, and some of the smaller islands [11,12]—although the occupation of smaller islands may be ephemeral [13,14]. The ability of wolves to disperse is widely documented, including over long distances and across a variety of habitats [15,16]. Wolves are even across water bodies, as they are capable of swimming and using smaller islands as stepping stones, which facilitates the colonization of islands [11,14,17,18,19].

Island populations may be susceptible to reductions in survival and persistence due to low genetic diversity, small population size, and limited gene flow from other populations [20], as has been demonstrated in canids (*Canis lupus* and *Urocyon littoralis)* [21,22]. Limited connectivity to other populations can increase breeding among closely related individuals, and the signatures of inbreeding have been detected in the islands of southeast Alaska [23,24]. Increased levels of inbreeding decreases heterozygosity and standing genetic variation, which hinders the capacity of populations to adapt to changes in the environment and thus may reduce their viability [25,26]. Therefore, a better understanding of wolf dispersal patterns among islands will help predict wolf population dynamics and persistence.

Wolves are structured into territorial social groups, the elemental unit of which is a breeding pair and their offspring [27]. Most non-breeding wolves of both sexes disperse from their natal territory to establish a new territory or join an existing pack and find mating opportunities and available resources [27]. Wolves that settle into a territory and mate then gain resident status, and their spatial patterns become constricted to the pack home range. Whereas dispersal is considered a permanent movement away from the resident home range to a nonadjacent area, wolves may also make extraterritorial movements outside of their home range which may be temporary or exploratory in nature and may precede permanent dispersal [15,28]. In addition, individual wolves may switch between resident and dispersal states during their lifetime [5,27], as secondary dispersal may be triggered by searching for an opportunity to breed if a mate was not found in the first dispersal destination [29].

Wolves in southeast Alaska have been the focus of conservation and management concerns since the 1990s, resulting in three petitions for listing as threatened under the Endangered Species Act, the most recent in 2020. These petitions have focused on Game Management Unit 2 (GMU 2), which encompasses Prince of Wales Island and the surrounding island complex (9025 km^2^; Figure 1) due to the high levels of wolf harvest and extensive old-growth forest logging that occur in this region [5,30]. In 2023, the U.S. Fish and Wildlife Service determined that listing was not warranted at that time [31]; however, GMU 2 was identified as the area with the lowest resiliency under current conditions, and under future threat conditions, it was projected to become functionally extirpated under the scenario of high wolf harvest (i.e., harvest season lasts ≥40 days) and high timber extraction (i.e., maximum harvest of old-growth forest) [31].

Although Prince of Wales Island makes up most of the landmass in GMU 2 (6670 km^2^, 74% of total area), there are 25 islands in the Prince of Wales Island complex larger than 5 km^2^, ranging from 5 to 661 km^2^ [32]. Wolves occur on these islands as evidenced by harvest locations, trail camera photos, and visual observations. To provide information for determining annual wolf harvest levels, the Alaska Department of Fish and Game (ADF&G) has conducted annual wolf abundance estimates in GMU 2 since 2013. Wolf densities are estimated using spatial capture–recapture and individual wolf genotypes from noninvasively collected hair samples [33]. Although the area of analysis has expanded over the course of the monitoring project, most of the sampling sites (136 out of 145) have been located on Prince of Wales Island apart from nine sites monitored sporadically on adjacent Sukkwan (169 km^2^) and Goat (17 km^2^) islands (Figure 1). Therefore, it is possible that the unsampled islands of GMU 2 harbor unaccounted variation in wolf densities and population dynamics and could provide an avenue for a source–sink relationship (i.e., differential population growth among patches in the network) with wolves on Prince of Wales Island. The ability of wolves to travel among islands would influence how quickly unoccupied territories may be recolonized after resident wolves have been harvested. In addition, inter-island dispersal patterns leading to reproduction and therefore gene flow are key factors influencing the dynamics of wolf packs on the GMU 2 islands [31]. 

Previous research in GMU 2 with radio and GPS collared wolves revealed little inter-island dispersal [34,35], and none of the collared wolves have been documented traveling between Prince of Wales Island and other GMUs or geographical areas in the Alexander Archipelago. Despite these efforts to characterize wolf dispersal patterns in southeast Alaska, a comprehensive understanding has been hindered due to the limited number of GPS collared wolves, and previous difficulties resighting and relocating VHF (very high frequency) collared wolves, especially when wolves disperse out of their origin GMU [36]. Noninvasive genetic methods used to monitor wolves have been increasingly applied over the past decade and used successfully to identify wolf packs, document dispersal [37,38,39,40,41], and use family group assignment to inform density estimates over large spatial scales [42,43]. In this research, we used DNA data from GMU 2 wolves collected during 2012–2021 in combination with location data from 13 GPS-collared wolves as part of our long-term monitoring efforts to provide insight into pack dynamics. The primary objectives were to identify dispersers and quantify straight-line dispersal distance, annual dispersal rates, and assess differences in dispersal rates and distances among male and females. In particular, we wanted to assess patterns of dispersal and the genetic structure of wolves across water bodies and among the islands of the Prince of Wales Island complex. Finally, we evaluated the association between wolf densities, harvest densities, and wolf dispersal. 

## 2. Material and Methods

### 2.1. Study Area

This study was conducted in GMU 2, which consists of Prince of Wales Island, the largest island in the southern portion of the Alexander Archipelago in southeast Alaska, and a complex of hundreds of adjacent smaller islands (Figure 1). This area contains rugged mountains with elevations ≤ 1160 m, an extensive coastline, and large tracts of temperate rain forests dominated by Sitka spruce (*Picea sitchensis*), Western hemlock (*Tsuga heterophylla*), Western red cedar (*Thuja plicata*), and yellow cedar (*Callitropsis nootkatensis*) at elevations below 600 m. Wolves in this area predominantly prey on Sitka black-tailed deer (*Odocoileus hemionus*), and important alternate prey include beaver (*Castor canadensis*), black bears (*Ursus americanus*), salmon (*Oncorhynchus* spp.), and mustelids [43,44,45].

Annual wolf density in GMU 2 ranged from 10 to 44 wolves/1000 km^2^ during 2013–2021 [33,46] (Appendix A). Human-caused wolf mortality during most annual hunting and trapping seasons has ranged from moderate to high and has influenced wolf densities [5,33] (Appendix A). Reported wolf harvest was high during the 1990s (average annual harvest n = 93, range 65–130), moderate from 2001 to 2014 following the implementation of a wolf harvest limit of 30% of the fall population estimate (average annual harvest: n = 49, range 20–89), and relatively low during 2015–2018 due to a reduction in the wolf harvest limit to 20% (average annual harvest: n = 35, range 7–61). In 2019, a new wolf management strategy established wolf population objectives in place of harvest limits, with the length of the harvest season as the mechanism for limiting or increasing the number of wolves harvested to meet the current population objective (150–200; regulation 5 AAC 92.008(1); [47]). After the 2-month 2019–2020 season, 164 wolves were reported as harvested in GMU 2, reinitiating discussion over sustainable levels of wolf mortality and triggering shorter seasons in the subsequent years, which resulted in reduced harvest during 2020 (n = 68) and 2021 (n = 66). 

### 2.2. Data Collection

Individual wolves were identified via genotyping DNA extracted from hair, muscle, blood, and scat samples collected both noninvasively and from harvested and captured wolves. Individual wolf relocation histories were compiled using spatially and temporally referenced genotypes identified from monitoring and harvest, wolf captures, and GPS collar location data. 

During the annual fall wolf monitoring effort (late September–mid-December 2012–2021), hair samples were collected weekly at an array of lured hair snares [33]. We used sterilized tweezers to pluck hair from snares, stored the hair in labeled coin envelopes, and dried at room temperature. During 2015–2018, we also collected wolf scat and shed hair at 13 active den sites within the home ranges of 9 wolf packs in GMU 2 and in the Snow Pass Islands [48]. Although the Snow Pass islands are administratively assigned to adjacent Game Management Unit 3, this area was included in our sample collection efforts because of the proximity to Prince of Wales Island and suspected population connectivity (Figure 1). Scats were placed in paper bags and stored in plastic bins or resealable plastic bags with silica gel for desiccation, and hair samples were stored in labeled coin envelopes at room temperature.

We captured wolves during 2012–2016, using the techniques previously described [49]. Briefly, we used modified or padded leghold traps set along the road system with commercially produced lures and canid urine used as attractants. Capture and handling procedures conformed to guidelines established by the ADF&G Animal Care and Use Committee (ACUC #2012–028 and #2014–15) and the American Society of Mammalogists [50]. We fitted each captured wolf with a spread-spectrum, Global Positioning System (GPS) radio collar (Mod 4500, Telonics, Inc., Mesa, AZ, USA). We immobilized restrained wolves with either tiletamine HCl and zolazepam HCl or a combination of ketamine and medetomidine. Blood was drawn by venipuncture with a needle and syringe for individual genotyping. 

Skin tissue samples were collected from harvested wolves during the mandatory sealing process. Locations of harvested wolves were georeferenced to the centroid of their reported uniform coding unit (UCU), which are specific areas within minor drainages of the GMU (mean area = 179.7 km^2^, SE = 32.3 km^2^), or to a more specific location if the description was available in the sealing records. Annual harvest records were also recorded and classified by the UCU to estimate annual harvest density. 

### 2.3. Genotyping

DNA extractions, genotyping, and sex identification of wolf samples were conducted at the National Genomics Center for Wildlife and Fish Conservation and previously described [33]. In brief, DNA was extracted from samples using standard protocols for tissues (DNeasy Blood & Tissue kit; Qiagen, Valencia, CA, USA). We modified the procedure by using overnight incubation in buffer ATL and Proteinase K on a rocker or in a rotating oven at 56 °C, a 70 °C incubation for 10 min after adding buffer AL, and a final elution using 100 µL buffer AE warmed to 70 °C. We analyzed hair and skin DNA extractions for individual identification using a panel of 15 variable microsatellite loci: cph5 [51]; fh2001, fh2010, fh2054, fh2079, fh2088, fh2096, fh2137, fh2140, fh2161, fh2548 [52], Pez17 [53]; c20.253 [54], VWF [55], and AHT130 [56]. Samples that amplified with 3 or more alleles at a single locus or that failed to be genotyped at 7 or more loci were discarded. In cases where a sample contained DNA from more than one individual, we implemented a single-hair DNA extraction protocol, consisting of selecting four hairs with follicles from different locations in the hair clump and performing a separate DNA extraction on each hair with a minimum of 2 re-extractions for each hair [33]. We used vertebrate primers to amplify a 360 bp 16S rRNA region of the mitochondrial genome, performed Sanger sequencing, and used NCBI BLAST to distinguish canid samples from non-target species. The sex of individual wolves was identified using the canid SRY marker [57].

After genotyping, we used DROPOUT v. 2.3 [58] and AlleleMatch [59] in R (v. 4.0.3, R Core Team, Vienna, Austria) to highlight individuals with incomplete loci matches and to identify allele scoring or data entry errors. We checked for homozygote excess due to null alleles and identified possible scoring errors using MicroChecker v. 2.2.3 [60]. We used GenAlEx v. 6.503 [61] to evaluate allele frequencies, verify low-frequency alleles and generate a principal coordinates analysis (PCoA) graph to determine if genotyped samples corresponded to reference populations of dogs or wolves. We also performed assignment tests using GeneClass2 [62] on all hair and scat samples and the same reference samples to distinguish if the samples originated from dogs or wolves. The 15 loci gave a cumulative probability of individual identity and probability of identity giving siblings as 2.17 × 10^−10^ and 6.074 × 10^−5^, respectively. We calculated the probability of identity (*P*_(*ID*)_) and probability of identity for siblings (*P*_(*ID*)sibs_) using GenAlEx 6.5 [61]. We tested for deviation from Hardy–Weinberg proportions and linkage disequilibrium using GENEPOP 4.7 [63] within the POW wolf population and between pairs of loci using the Markov chain Monte Carlo approximation of Fisher’s exact test and a simulated exact test, respectively. We ran 10,000 dememorizations, 100 batches and 5000 iterations, and applied a Bonferroni-corrected alpha level of 0.05 for multiple comparisons. We measured the level of genetic diversity and variation by calculating the mean number of alleles per locus (*A*) as well as the mean observed and expected heterozygosities (*H_o_* and *H_e_*) using GenAlEx. We calculated rarefied allelic richness (*AR*) richness corrected for sample size differences using hp-rare 1.0 [64].

### 2.4. Group Assignment

We assessed pack structure and estimated individual maternal and paternal relationships using a maximum-likelihood framework implemented in CERVUS 3.0.7 [65], which allows the estimation of critical values of the difference in log-likelihood (LOD) values between putative parents. Parentage simulations generated 10,000 offspring with 407 candidate mothers and 423 candidate fathers (all males and females available in the data set), assuming 60% of the population was sampled and 1% of the loci were mistyped. We required critical trio (including both putative parents and offspring) LOD scores at the strict (95%) confidence rate to assign an individual to a family group. 

We assessed pairwise relatedness among all genotyped wolves using ML-RELATE [66]. A wolf was assigned to a family group if the majority of the individuals genotyped within the same pack territory within the same year had relatedness levels *r* ≥ 0.5 indicating parent–offspring (PO) or full-sibling (FS) relationships [38,40]. 

We estimated the number of genetic clusters (*K*) in the population based on previous work indicating that closely related family groups will share similar allele frequencies and may be used for pack assignment [38,40,67]. We used a Bayesian cluster procedure implemented in STRUCTURE v2.3.3 [68] to assess the average proportion of membership (*q*) assigned to clusters. We used a general admixture model with no location prior and correlated allele frequencies. We performed the analysis 10 times independently after a burn-in of 100,000 and used 500,000 MCMC repetitions where *K* = 1–25. We determined the value of *K* first by calculating the maximum likelihood value (ln[Pr(X|K)]) and second with ∆K [69] and determined individual *q* values with STRUCTURE HARVESTER 0.6.94 [70]. For group assignment, individuals required a *q* > 0.7.

### 2.5. Wolf Dispersal

We assigned wolves to the following categories: “resident”, “disperser”, and “uncategorized” to reflect social status inferred from recapture and genetic data. We used previously defined home ranges from GPS-collared wolves [49] to identify pack territories. In regions of GMU 2 where GPS-collared wolves were not distributed, we determined the occurrence of a wolf pack when we detected at least 2 resident wolves in a common area such as a minor drainage [38]. Wolves were considered resident when they met the conditions of at least one of the following criteria: (1) at least one relocation was at an active den [71], (2) at least 2 relocations were within a pack territory [72], or (3) at least one relocation was within a pack territory and 2 of the pedigree analyses methods (described in Section 2.4) provided corroborating evidence of group assignment assigned wolves to the group inhabiting the pack territory.

Dispersers were categorized from the GPS locations of collared wolves, or from genetic recaptures of an individual in one wolf pack territory, and then subsequent recaptures in another wolf pack territory [72]. Wolves were also categorized as dispersers if they were recaptured in a wolf pack territory but demonstrated strong evidence of family assignment to a different wolf pack with at least 2 of the group assignment methods providing corroborating evidence [38,40]. Wolves were classified as uncategorized when only detected once and lacking strong assignment to any family from pedigree analyses.

We estimated dispersal distance as the straight-line distance between the last relocation of an individual and the first relocation in a different home range [41]. When wolves lacked relocations in their origin pack territory but genetically assigned to the origin pack, we used the location of the wolf pack den site as the first location for wolves that dispersed to a different wolf pack [38]. The estimated dispersal distances are minimum distances and do not represent the travel path or total distance of dispersal. We estimated annual dispersal rate as the proportion of wolves that dispersed out of the total number of wolves identified each year [72]. We determined the sex ratios of all sampled wolves collectively and dispersing wolves using molecular methods and tested for differences in the dispersal rate between males and females using a two-tailed unpaired two sample *t*-test. We tested for differences in minimum dispersal distances between males and females using a Mann–Whitney U-test with a 0.05 significance level. We categorized the fate of dispersing wolves as (1) died while dispersing (evidence wolf died from harvest records and associated DNA samples during sealing or from monitoring GPS collared wolves, ≤1 relocation in a territory outside of their natal territory, and died during the same biological year as wolf dispersed), and recorded cause of death, (2) joined or established a territory (>1 relocation in destination territory [72]), (3) did not settle (continued to move among wolf pack territories). If wolves that joined or established a territory or did not settle later died, we also recorded the cause of death.

### 2.6. Genetic Structure

We assessed the genetic structure of wolves on Prince of Wales Island in relation to the outer islands of GMU 2 using discriminant analysis of principal components (DAPC) [73]. The DAPC maximizes between-population genetic variation while minimizing within-group variation and allows a visual representation of genetic connectivity among groups [73]. We implemented the DAPC in program R using package adegenet 2.1.1 [74] in the R 4.2.3. environment (R Core Team, 2023). We defined groups based on assignment methods (described in Section 2.4) and did not include wolves that did not assign to a group (“uncategorized”). For visualization of the genetic structure and connectivity of wolves between the islands of GMU 2, we combined the wolf groups on Prince of Wales Island and plotted in relation to the wolves that were assigned to groups on the outer islands. We used cross-validation to determine the optimal number of principal components retained.

### 2.7. Wolf Harvest, Density, and Dispersal Analysis

We estimated annual wolf harvest density within the origin and destination territories of the dispersing wolves by dividing the number of wolves harvested in each biological year (May 1–April 30) in the corresponding UCUs by the total UCU area (Appendix A). We assessed differences between harvest densities in origin and destination territories using linear regression and with graphical exploration in the program R. Because most of the harvest occurs after the intensive sampling season (October–early December), and we wanted to assess the relationship between harvest intensity and dispersal patterns, we used the harvest density of the prior year for the origin and destination UCUs. For comparison purposes, only wolves that had identified “resident” locations within one year prior to the dispersal event were included.

We assessed the relationship between annual wolf density and annual dispersal rates using linear regression in the program R with wolf dispersal as the response variable. We also assessed wolf dispersal the year following a given wolf density.

## 3. Results

### 3.1. Genotyping

We collected 7891 snagged hair samples from annual monitoring efforts during 2012–2021 and 322 wolf hair samples and 227 wolf scat samples (114 adult scats and 113 pup scats) at active den sites during 2015–2018 [48]. We also collected samples from 14 captured wolves, 580 harvested wolves, and 2 road-killed wolves. We identified 1283 samples from non-target species including American black bear, n = 1283; beaver, n = 13; Sitka black-tailed deer, n = 4; American marten (*Martes americana*), n = 6; American mink (*Neovison vison*), n = 1; mouse (*Peromyscus* sp.), n = 1; and undetermined, n = 22. In addition, we identified 57 dogs (*Canis lupus familiaris*) from the *Canid* spp. PCoA and assignment tests and removed these from further analyses. After removing non-target species, we had 828 wolves remaining, which were genotyped at 15 microsatellite loci. We further removed 17 wolf genotypes obtained from harvested wolf samples (2 were duplicates of wolves harvested 5 years prior in a different GMU, and 15 wolves were assigned to other GMUs using PCoA analyses) to restrict analyses to wolves we could confidently assign to GMU 2. Thus, our final data set consisted of 811 wolves (394 females, 412 males, and 5 individuals the sex of which could not be determined).

The genotyping success rate (the percentage of samples screened as *Canid* spp. that successfully amplified and passed quality control steps) of identifying individual wolves varied among sample type. The genotyping success rate of snared hair collected during annual monitoring efforts was 51% for standard extracts (2448 out of 4781 samples) and 24% for single-hair extracts (191 out of 793 samples). The genotyping success rate for samples collected at active den sites was 39% for shed hair (124 out of 321 samples) and 53% for all scat samples (114 out of 214 samples). Muscle and skin tissues from harvested wolves and blood from captured wolves had a 99% genotyping success rate (572 out of 580 samples). All loci were polymorphic with 3–10 alleles across loci (*A* = 7, *AR* = 3.08). *H_o_* and *H_e_* were 0.544 and 0.561, respectively.

### 3.2. Group Assignment and Wolf Dispersal

We detected 27 wolf groups (Figure 2a) and could assign 64% of individuals to the resident category (n = 518), while 18% we could not assign to a group (uncategorized; n = 149), and 18% (n = 144) met the criteria for assigning to a group and were then subsequently relocated in another territory and were classified as dispersers. Annual dispersal rates ranged from 9 to 23% (Table 1). Of the dispersing wolves, 66 were females and 78 were males, and there was no significant difference in dispersal rate between sexes (*t*-test: t = 1.19, df =9, *p*-value = 0.262, 2-tailed). The minimum dispersal distance ranged between 6 and 138 km (mean = 41.9 km, SD = 23.7 km), and males had longer dispersal distances (mean = 44.8 km, SD = 23.3 km) than females (mean = 38.5 km, SD = 23.9 km) determined by the Mann–Whitney U test (W = 2048, *p*-value = 0.035), although the effect size between male and female dispersal distances was small. Overall, 39 of the 144 dispersing wolves (21 females and 18 males) traveled either between Prince of Wales Island and the outer islands or among the outer islands (Figure 2b). The minimum distance for these inter-island dispersal events, including travel across water, was similar (mean = 40.5 km, SD = 3.5 km) to the dispersal distances occurring exclusively on Prince of Wales Island, and there was no difference between male and female inter-island dispersal distances (W = 170, *p*-value = 0.612). During our study period, 18 wolves dispersed from Prince of Wales Island to outer islands, 13 wolves dispersed from outer islands to Prince of Wales Island, and 8 wolves dispersed among the outer islands indicating connectivity throughout the island complex. The remaining 105 wolves dispersed within Prince of Wales Island.

Of the dispersing wolves, 107 died (51 females and 56 males), and 26 of these wolves died after dispersing to one of the outer islands. The majority died from trapping or hunting during the annual harvest season (n = 104), one wolf died from aspiration pneumonitis during capture handling, and two wolves died from intraspecific strife either during dispersal movements or while defending their destination home range where they had settled. However, because DNA samples were obtained from harvested wolves during the sealing process, our relocation data reflected legally harvested wolves, as wolves that died from unreported human-caused mortality or from natural causes would be difficult to detect unless included in our GPS-collared wolf sample. The majority (n = 81) of dispersing wolves died before they were able to settle. Twenty-four of the dispersing wolves settled in a new territory (9 females and 15 males), 10 of which died in this territory. The duration of occupancy of settled wolves before dying ranged from 53 to 998 days (mean = 529, SD = 346). We could not determine if 16 wolves were in a dispersing state or had settled when they died due to a temporal gap in relocation data.

### 3.3. Genetic Structure

Our discriminant analysis of principal components (DAPC) showed connectivity between the wolves that were assigned to groups on Prince of Wales Island (n = 537) and wolves (n = 126) that were assigned to groups on 10 of the outer islands (Dall, Heceta, Kosciusko, Long, Noyes, San Juan de Bautista, Snow Pass complex, Suemez, Sukkwan, and Tuxekan; Figure 3). Wolf groups on Prince of Wales Island overlapped when groups were defined separately and not combined. The Prince of Wales Island wolves as a group were central in our DAPC analyses, and all outer wolf groups overlapped with this cluster. Island groups radiated off the central group in subclusters with Noyes, San Juan de Bautista, and Heceta along the right side of the *x*-axis, Long, Dall, Kosciusko, and Snow Pass on the left side of the *x*-axis, and Suemez on the upper *y*-axis (Figure 3). Tuxekan and Sukkwan Island wolves had the highest degree of overlap, completely overlaying the distribution of Prince of Wales Island wolves. Suemez Island wolves had the least amount of overlap but showed connectivity to Prince of Wales Island and Sukkwan Island. Island subclusters conformed to the geographic proximity of islands excepts for the Snow Pass Island complex and Kosciusko (on the northeast and northwest side of Prince of Wales Island, respectively), which grouped with Dall and Long (located in the southwest sector of the GMU 2 complex). We retained 40 principal components, as suggested by the cross-validation, but tested a range of values of retained principal components (from 20 to 50) to assess any differences in genetic structure and found none.

### 3.4. Wolf Harvest, Density, and Dispersal Analysis

At low levels of harvest density (<0.01 wolves/km^2^), wolves tended to disperse to areas that had higher harvest densities in the prior year (Figure 4). However, this trend reversed as the harvest rate density increased in the origin territory (i.e., the area the dispersing wolf was last known to be a resident). At higher harvest rate densities (>0.01 wolves/km^2^), wolves tended to disperse to areas with lower harvest rate densities than the territory of their origin. There was a significant, negative linear relationship between harvest density in the disperser’s territory of origin and the territory it dispersed to (*R*^2^ = 0.29, F(1109) = 46.91, *p* < 0.001). Overall, wolf dispersal rates had a positive relationship with wolf density during the preceding year, although this was not significant at the 0.05 level (*R*^2^ = 0.42, F(1,6) = 4.362, *p* < 0.082).

## 4. Discussion

Wolf genetic data collected over a decade of monitoring efforts noninvasively opportunistically from harvested wolves revealed a higher degree of population connectivity across the islands of GMU 2 than previously known. Our results demonstrate dispersal occurring regularly and throughout large portions of GMU 2 including across water bodies to connect the outer islands with Prince of Wales Island. Dispersal occurred both to and from small islands to the larger Prince of Wales Island, indicating bidirectionality of movement as opposed to a consistent source–sink dynamic. Additionally, we revealed dispersal among several of the outer islands which showed connectivity among island clusters, which are results supported by our genetic structure analyses displaying an overlap of some outer island wolf groups (Figure 3).

Wolves were captured intermittently during the past 3 decades and instrumented with VHF (during 1993–1995, 1999–2004) or GPS (during 2012–2016) collars mainly on Prince of Wales Island but also on three of the outer islands (Heceta (n = 9), Kosciusko (n = 3) and Tuxekan (n = 1); [34]). During these study periods, four wolves traveled between an outer island and Prince of Wales Island, in some cases involving multiple swims among stepping-stone islands, all shorter than 0.5 km [34]. Three of these wolves dispersed from one of the outer islands (Tuxekan: n = 1, Kosciusko: n = 2) to Prince of Wales Island (although one of these continued to disperse, finally settling on Dall Island), and one wolf originating from Prince of Wales Island was killed on one of the small adjacent islands (Signal Island, <1 km^2^). Thus, 4 out of 68 collared wolves dispersed among five islands over the course of 16 monitored years (6%). In this study, we detected 39 wolves dispersing among 18 islands in 10 years (5% of genotyped wolves) with annual inter-island dispersal rates ranging from 1 to 7%. The benefit of using genetic information during the current study was an augmentation of the number of wolves we could monitor annually, an expanded geographic area that we could effectively monitor, and an extended study period which has previously not been possible in this area relying solely on collared wolves.

The occupation of small islands by wolves and the duration of their tenure on these islands is presumed to be ephemeral based on past research and local knowledge in the coastal geographies of southeast Alaska and British Columbia [13,14]. In GMU 2, it has been suggested that wolves only consistently occupy the three largest Islands in the complex (Prince of Wales Island, Kosciusko (447 km^2^), and Dall (661 km^2^)). Moreover, when accounting for the carrying capacity of their main prey (Sitka black-tailed deer), modeled wolf–deer interactions, and in the absence of immigration, only Prince of Wales Island meets the predicted area requirement of 2000–3000 km^2^ to ensure a persistent wolf population [34]. Examples such as the experimental transplant of wolves to Coronation Island (73 km^2^) and their eventual demise [13] have led to the conclusion that wolves will persist for longer time frames on larger islands due to there being a more diverse resource base [75] and more stable predator–prey interactions [3], and on less isolated islands, allowing for recolonization of terrestrial prey [8]. Notwithstanding, the occupation of wolves on islands as small as 0.7 km^2^ (Moore Island [8,11]) during a 5-year study period, and Chatham and Discovery Islands (1.9 km^2^) in British Columbia for 7 years [14], and on islands ≤ 1.34 km^2^ in GMU 2 [76] during a 5-year study period show wolves use these landforms for longer time frames. The 10-year residency of a wolf pack exclusively on Pleasant Island (50 km^2^) in southeast Alaska even after the resident deer population was depleted by predation [19] indicates that islands may provide other benefits for wolves that compensate for what they lack in area. These benefits include access to marine resources that are predictable, energetically efficient to acquire, and may lead to increased abundance of the predator [77,78,79]. Additionally, the geography of an island results in a well-defined home range requiring less effort to patrol and defend [27] and potentially less exposure to human-caused mortality.

Wolf packs are also known to occupy multiple neighboring islands or to include both islands and areas of adjacent coastal mainland such as peninsulas into their home ranges [18,19,75]. The temporary use of a series of islands, incorporation of small islands into the home range area of Prince of Wales Island wolves, or short-term (i.e., a couple of years) exclusive residency on a small island may also occur [80]. Traditional ecological knowledge gained from Indigenous residents of Prince of Wales Island provides similar perspectives—that wolves may move from one island to another, either in a pattern of temporary expeditions to a neighboring island but always returning to the original island, or a pattern of wolf packs serially recolonizing new islands after exhausting the local deer population through predation [81]. Residents also believed that typically one pack of wolves inhabit each island, although some of the larger islands such as Dall may contain multiple packs. Importantly, these ethnological reports describe large pack sizes on islands and a quick rebound of wolf reoccupation of islands after intensive harvest at that site the previous year [81].

Our results echoed these observations, showing wolf dispersal among several of the outer islands ranging in size from 22 to 661 km^2^ (mean = 212 km^2^) corroborating the ability of wolves to spend their life cycle inhabiting smaller islands. Interestingly, wolves did not disperse to the islands immediately adjacent to their home territory during these inter-island movements (mean distance = 39 km, range = 25–67 km), suggesting that dispersing wolves swum between other islands, connecting them in their route. In addition to measuring individual dispersal events among the outer islands, our genetic structure analyses also showed a strong overlap of some island groups such as Dall and Long Islands as well as Noyes and San Juan de Bautista Islands, suggesting that wolves may include multiple islands within their home range. We found that the majority of dispersal events resulting in movements off Prince of Wales Island were to relatively small islands 22–176 km^2^ instead of one of the larger islands in the archipelago. Wolves dispersed from Prince of Wales Island to both the island immediately adjacent to their home territory (56% of dispersers) and to non-adjacent islands (44% of dispersers), in some cases covering long distances (mean distance = 41 km, range = 6–112 km) and requiring either travel across the main Prince of Wales Island or by connecting a series of islands to reach their destination.

These small islands appear to play an important role in the population dynamics of GMU 2 wolves, not only in providing stopovers during travel to facilitate population connectivity across water bodies, but also as a refuge from which dispersers may radiate from. Even if only used temporality, small islands may thus serve as a source to recolonize other island territories that may be unoccupied, have low densities of wolves, or have vacant breeding positions. Patterns of human harvest of wolves may exacerbate the observed dispersal dynamic in GMU 2 by opening such vacancies. Certain island populations of wolves in GMU 2 are targeted by resident trappers with the objective of creating havens for deer, and by extension, for resident deer hunters; therefore, wolves can be depleted on islands due to localized harvest [81]. Additionally, the serial or cyclical occupation of islands by wolves may create cycles of exploitation of the local prey (i.e., deer) followed by periods of prey population rebound and recolonization after the wolves have vacated for a different island [76]. As has been demonstrated in continental systems with more continuous landscapes, wolf subpopulations occur at various densities influenced by prey availability and are connected by dispersal to form a metapopulation [29,82,83,84]. The ephemeral nature of wolf occupation of small islands, facilitated by their recognized ability to swim, may in fact be the key to wolf persistence, as it allows them to recolonize from across the network of islands after local extinction or reduction in population density.

Dispersal may be effective in connecting discrete patches across a metapopulation demographically, but for genetic connectivity to be established, reproduction and gene flow must occur. Therefore, dispersing wolves must survive long enough to join or form a pack and reproduce. Our results show high mortality among dispersing wolves, with 74% dying, mostly from human-caused mortality. Previous research in GMU 2 also identified a low mean annual survival of nonresident wolves, which included both dispersers and extraterritorial wolves (0.34, SE = 0.17), in comparison to resident wolves (0.65, SE = 0.17; [5]). When dispersing wolves were considered separately, Person and Russell reported 16% annual survival with hunting and trapping the dominant cause of death [5]. Higher rates of mortality in dispersing wolves than resident wolves have been reported in other systems where human harvest is common [4,85,86]. A meta-analysis of global wolf populations concluded that human-caused mortality of dispersing wolves also reduced the dispersal distance, duration, and success of establishing a home range or joining an existing pack [16]. In our current study, we found that most dispersing wolves (56%) died before settling in a new territory, whereas 17% successfully settled. We recognize that cautious interpretation of our results is necessary, as natural mortality or wolves killed illegally (reported as 9% and 29% of the study population, respectively; [5]) would be underrepresented. Still, our low rates of dispersers settling due to human-caused mortality is comparable to rates reported for GMU 2 wolves previously (19%) based on GPS-collared wolves [5]. The rate of dispersers successfully joining or establishing a pack in GMU 2 is substantially lower than that found in interior Alaska (42%) [87], the Italian Apennine Mountains (59%) [88], the Italian Alps (51%) [41]), or globally (77%) [16].

Most dispersal mortalities occurred during the same biological year that dispersal from the home range was initiated, and except for one wolf that died from intraspecific strife, and one that was legally hunted, all died from trapping (74%) during the annual harvest season. The trapping season has historically begun on December 1 under State of Alaska regulations, and November 15 under Federal subsistence regulations, but the length of the season has varied over time. After the harvest quota was reduced to 20% of the estimated wolf population during 2015–2018, more wolves were harvested earlier in the season (i.e., December–January), whereas before this change in regulation, wolves were more commonly harvested later in the winter and early spring (i.e., February–March). In 2019, the state trapping season opening date shifted to align with the federal season beginning November 15, and management switched from a quota system to a season-length based system to achieve a wolf population objective of 150–200 wolves (regulation 5 AAC 92.008(1), [47]). Most wolf dispersal occurs during late fall, winter, and spring [16,27,84]; therefore, the harvest season coincides with the time of the year wolves most commonly disperse, and it likely led to the high harvest rates of dispersing wolves in our study area and the low number of documented settlers. We do not know the age structure of the dispersing wolves in our sampled population, but based on past research in our study area and elsewhere, most dispersers are nonbreeding wolves ≥ 2 years old [5,27,87,89]. As these dispersing wolves are seeking breeding opportunities, and wolves may breed by 22 months [89,90], death before settling of these dispersing wolves curtails their potential to breed and reduces gene flow potential among packs and subpopulations. However, as breeding season occurs in February through early March, dispersing wolves that either survive the annual harvest in early winter or disperse later in winter would have a better chance of settling and ultimately breeding.

While high human-caused mortality may have reduced disperser breeding success, it also may have contributed to dispersal patterns by creating breeding opportunities for subordinate wolves. In our study, we found that when localized wolf harvest occurred at low levels, wolves dispersed to areas of high harvest density, suggesting that human-caused mortality provided vacant territories for dispersing wolves to occupy or available breeding positions in packs. Areas of low harvest density in our study area (<0.01 wolves/km^2^) generally experienced <2 wolf removals per UCU during the harvest season. Wolves originating from these areas dispersed into areas of higher harvest densities, which had experienced 2–27 wolf removals during the prior harvest season. In areas of high harvest, loss of breeders can create breeding opportunities for unrelated wolves dispersing from adjacent packs, which maintains stability of pack occupancy despite high removal rates [91,92,93,94]. Dispersers are recognized as providing a key role in wolf population dynamics by replacing breeding pack members that have died [27,89,95,96], so the ability of dispersing wolves to successfully establish a territory or integrate into an existing pack may be facilitated in areas of high harvest if wolves can avoid being harvested themselves. In our study area, we also found that when localized wolf harvest rates surpassed the threshold of >0.01 wolves/km^2^, wolves originating from these high harvest areas dispersed to an area with lower harvest density. The pattern of dispersing to areas with lower harvest densities was especially pronounced for wolves originating in territories with harvest densities in the 0.02–0.03 wolves/km^2^ range, which equated to six to eight wolves removed from a UCU during a season. This switch in dispersal patterns at the >0.01 wolves/km^2^ threshold of harvest density could be due to a variety of factors, including potential effects of pack dissolution, avoidance of anthropogenic disturbance, or other unmeasured factors such as more abundant prey in the territories the wolves dispersed to, but overall, it demonstrates flexibility in the behavior and strategies of wolves in areas with variable harvest levels.

The overall percentage of dispersers in the GMU 2 population during the study period was 18% and annual rates ranged from 9 to 23%. This is lower than the annual rate of dispersal (39%) reported by Person and Russell [5] and lower than the proportion of nonresident wolves (29%) in the GMU 2 population in the early 1990s (but dispersers and extraterritorial wolves were combined). Dispersal rates vary widely among areas studied with the long-term average in North American populations 10–40% [89] and may be influenced by wolf densities, human-caused mortality, prey availability, or intraspecific competition [29,84,89,97,98]. We found a weak positive relationship with wolf population dispersal rates and wolf density during our study period. Wolf densities are believed to have a strong influence on annual dispersal rates with the universal trend conforming to a bimodal pattern of high dispersal at high and low population density but low dispersal rates at intermediate values [16]. The dispersal rates of younger age classes (pups and yearlings) can increase when wolf density reaches the predicted carrying capacity based on local prey abundance [29]. Wolf densities in GMU 2 ranged broadly during the study period at least in part due to variable harvest levels and changes in harvest management; thus, continuing to track wolf densities and dispersal rates annually is recommended to further explore dispersal patterns and trends.

## 5. Conclusions

The use of long-term genetic data and recapture histories of individual wolves promoted a broader understanding of wolf dispersal patterns, including the demographic and genetic connectivity of small islands in GMU 2 with Prince of Wales Island. This spatial structure of wolf groups may influence the population dynamics of wolves in the archipelago, particularly in relation to their ability to compensate for localized high harvest. Dispersal allows for an overall increase in wolf abundance through expansion into vacant territories while maintaining densities in core population areas in a metapopulation framework [1]. The maintenance of core areas and the capacity for connection through dispersal is therefore a mechanism for population persistence in areas of high harvest [99]. The islands of GMU 2 are managed collectively under one harvest guideline, and local harvest densities are not consistent over time, suggesting a spatially and temporally patchy landscape of wolf removal and recolonization. The results of this study and other monitoring efforts in GMU 2 have indicated that the wolf population in recent years has been maintained at a relatively high density despite some years when high harvest occurred. In addition, due to the possibly of higher densities of wolves on islands, but with variable occupancy rates, islands may serve as a source of wolves to reoccupy areas that have experienced high harvest. Future work should investigate any variation in wolf densities on the outer islands of GMU 2 and determine how to incorporate this information into ongoing wolf monitoring efforts occurring mainly on Prince of Wales Island.

## Figures and Tables

**Figure 1 animals-14-00622-f001:**
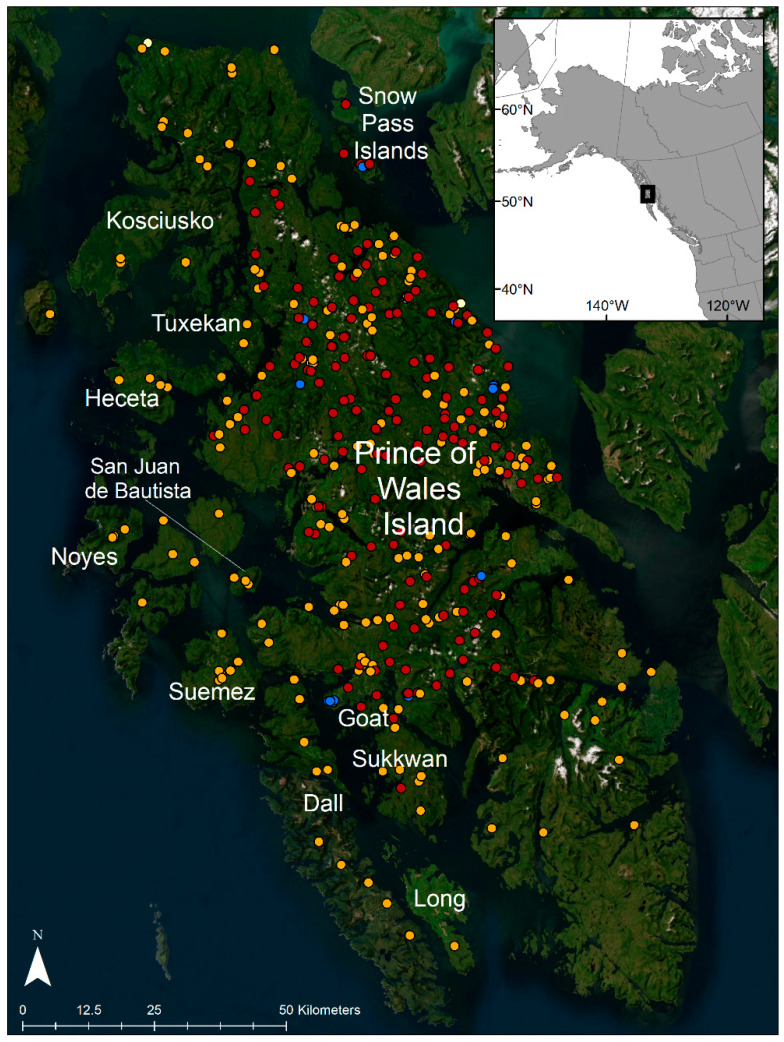
Wolf sample locations, GMU 2 (including Prince of Wales Island and the outer islands), AK, USA, 2012–2021. Samples from harvested wolves are shown as orange, hair samples are shown as red, scat samples are shown as blue, and blood samples are shown as white.

**Figure 2 animals-14-00622-f002:**
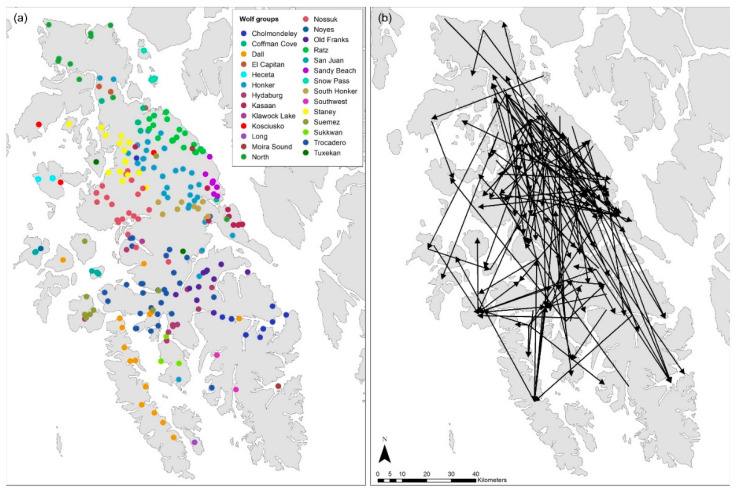
Wolf groups (**a**) and straight-line representations of dispersal distances (**b**), GMU 2 (including Prince of Wales Island and the outer Islands), AK, USA, 2012–2021.

**Figure 3 animals-14-00622-f003:**
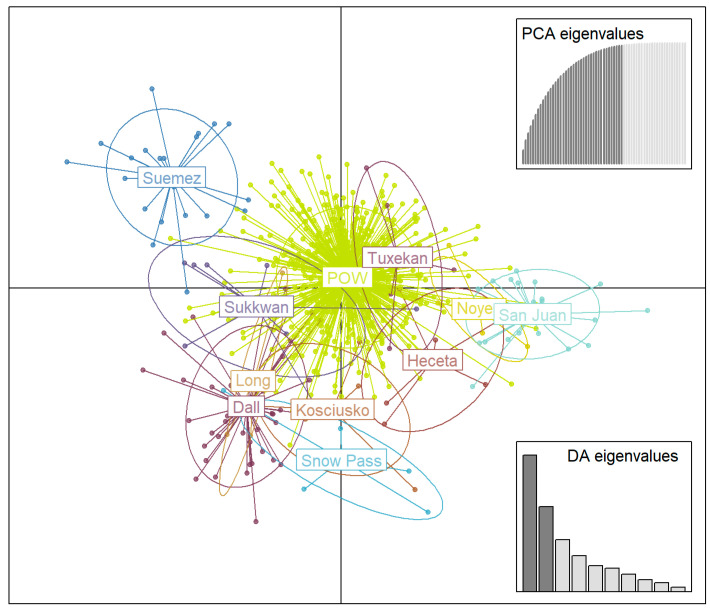
Results of discriminant analysis of principal components (DAPC) with number of axes retained = 40, for wolves on Prince of Wales Island and the outer Islands, AK, USA, 2012–2021.

**Figure 4 animals-14-00622-f004:**
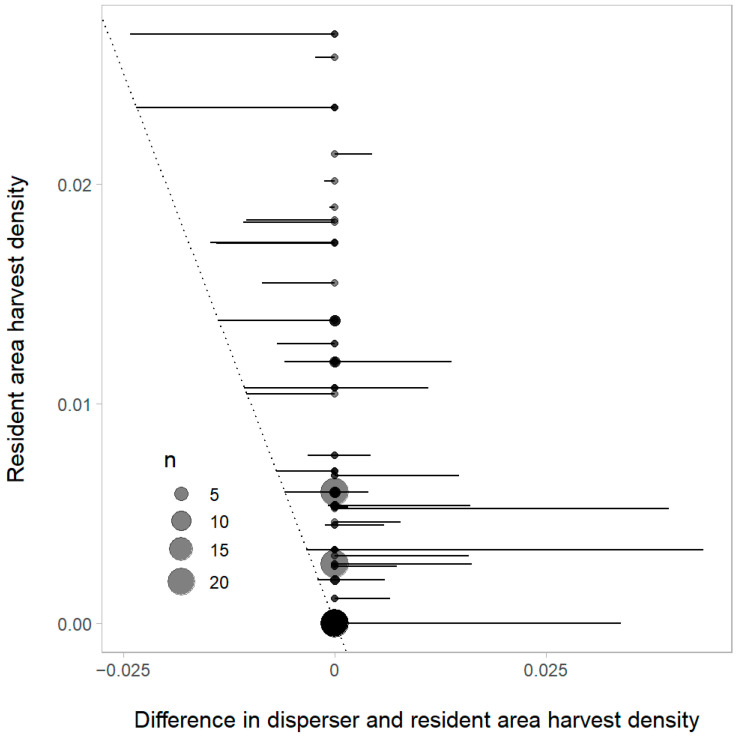
Difference between wolf harvest densities in the areas a wolf dispersed to and the area the wolf was last known to be a resident in the year before sampling, GMU 2 (including Prince of Wales Island and the outer Islands), AK, USA, 2012–2021. Horizontal lines to the right of zero indicate a wolf dispersed to an area with a greater harvest density, lines to the left indicate dispersal to an area with a lower harvest density. The points indicate sample sizes, the horizontal lines are the difference in harvest density (dispersal destination area–pre-dispersal [resident] area), and the diagonal dashed line indicates an individual dispersed to an area with no harvest in the prior year. Points are scaled to sample size and are semi-transparent so that close values are visible.

**Table 1 animals-14-00622-t001:** Number of wolves identified from individual genotyping, number of dispersing wolves, dispersal rate, and the number of male and female dispersers by year in GMU 2 (including Prince of Wales Island and the outer Islands), AK, USA, 2012–2021.

Year	Wolves Identified	Dispersers	Dispersal Rate	Male	Female
2012	51	8	0.16	4	4
2013	63	9	0.14	5	4
2014	46	4	0.09	3	1
2015	38	6	0.16	5	1
2016	99	8	0.08	5	3
2017	125	25	0.20	14	11
2018	123	18	0.15	8	10
2019	231	34	0.15	15	19
2020	134	23	0.17	13	11
2021	105	24	0.23	12	11

## Data Availability

The data presented in this study are available on request from the corresponding author. The data are not publicly available due to agency mandates on restrictions of animal location and harvest information.

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
