# Peer review of "Patterns of Wolf Dispersal Respond to Harvest Density across an Island Complex"

_animals, 2024, doi:10.3390/ani14040622_

Round 1
Reviewer 1 Report
Comments and Suggestions for Authors
This is a well-written paper analysing long-term data on dispersal in wolves in a fragmented landscape. Data are derived from different sources. Statistical methods are solid and the discussion is based on the data. I have only a few comments specifically about some of the results that need some more detail (see below).
More detailed comments can be found below.
Introduction:
Lines 52-55: Add references to support this statement.
Lines 79-82: Add references to support this statement.
Lines 109-112: Add references.
Lines 124-126: Add references.
Lines 133-134: Explain what VHF stands for.
Lines 141-146: How does this data set differ from earlier studies, particularly when the same islands were sampled? You mention is further down in the discussion but it would be nice to read about it here to justify the study.
Methods:
Lines 155-158: have scientific names in italics.
Results:
Lines 353pp: It is unclear from the description, how these 828 wolf samples were selected.
Lines 379-381: What about the remaining 105 dispersing wolves?
Lines 397-399: How do you know this? When the wolf was in a different location than before then he had dispersed. But how can you know that he had not arrived in his final destination?
Fig. 4: Wolves originating from high harvest densities dispersed to lower harvest densities and vice versa. The question is how much this is affected by availability, i.e., how likely is it to find an in-range area that has higher harvest densities than an already high harvest area of origin? Likewise, how likely is it to find an in reach low harvest density area when originating from a low-density harvest area? It might be more likely to encounter higher density harvest areas when dispersing from low harvest areas. Also, might there be a spatial correlation between low and high harvest density areas? Are they clumped or evenly distributed?
Discussion:
Lines 484-486: Add references.
Reviewer 2 Report
Comments and Suggestions for Authors
Overall, this was a really interesting paper. I have made multiple comments on the .pdf itself so I won't re-write them all here.
There were some larger changes that I think needed to be addressed. First is the methods section. The collection methods were incredibly convoluted and hard to follow. I think you would really benefit from a "general" section that discussed the overall methods. As it stands, you don't really know what medium you're collecting until mid-way through your methods section. Also be sure that you're discussing WHY you used both scat and hair - did you analyze them the same way to look for the same information? How did you know that some of the samples were from pups when you weren't able to test for age in your analyses? Just make sure that scientists who may not be 100% familiar with your methods are able to understand the logic and the chronological order of your field collections. Also make sure you indicate if you were the only one who did the collecting, if you were in a team, or if you had samples donated (such as from the harvested animals).
The second large(ish) change I think would help the overall flow of your manuscript would be to include the figures when you mention them first. Right now, all your figures are extremely difficult to reference and when you discuss the x-axis and the y-axis in-text things quickly get confusing.
Finally, I think you need to discuss a bit more of the why. You mention some human harvesting / conflict but you don't discuss why knowing dispersal patterns would be important to know - just that they are important. I think you need a little bit of information either within your introduction or at the end of your discussion as to why this information is so important. Really tie the data and new knowledge into the "big-picture" of wolf conservation and human/wildlife conflicts.
Otherwise, I think your manuscript is well done and you clearly put in a ton of work. Please see the attached .pdf for grammatical / some contextual comments.

Grammar was typically fine but occasionally you ended up with some run-on sentences. Also really look at the commas you are using and whether or not you need them. Commas before "and" specifically - sometimes you put a comma before "and" when there are not two different independent clauses. Finally, really look at your tenses. You flipped back and forth a bit between each section on if you were reporting in past or present tense. Pick one and stick to it.
